# The Role of Hypomagnesemia in Cardiac Arrhythmias: A Clinical Perspective

**DOI:** 10.3390/biomedicines10102356

**Published:** 2022-09-21

**Authors:** Alina Gabriela Negru, Anda Pastorcici, Simina Crisan, Gabriel Cismaru, Florina Georgeta Popescu, Constantin Tudor Luca

**Affiliations:** 1Department of Cardiology, “Victor Babeş” University of Medicine and Pharmacy, 300041 Timisoara, Romania; 2Department of Cardiology, Institute of Cardiovascular Diseases, 300310 Timisoara, Romania; 3Rocordis Cardiology Hospital, 300174 Timisoara, Romania; 4Department of Internal Medicine, Cardiology-Rehabilitation, ‘Iuliu Haţieganu’ University of Medicine and Pharmacy, 400347 Cluj-Napoca, Romania; 5Department of Occupational Health, Victor Babeş University of Medicine and Pharmacy, 300041 Timisoara, Romania

**Keywords:** hypomagnesemia, supraventricular arrhythmia, torsade de pointes, ventricular arrhythmia, ventricular tachycardia, magnesium supplementation

## Abstract

The importance of magnesium (Mg^2+^), a micronutrient implicated in maintaining and establishing a normal heart rhythm, is still controversial. It is known that magnesium is the cofactor of 600 and the activator of another 200 enzymatic reactions in the human organism. Hypomagnesemia can be linked to many factors, causing disturbances in energy metabolism, ion channel exchanges, action potential alteration and myocardial cell instability, all mostly leading to ventricular arrhythmia. This review article focuses on identifying evidence-based implications of Mg^2+^ in cardiac arrhythmias. The main identified benefits of magnesemia correction are linked to controlling ventricular response in atrial fibrillation, decreasing the recurrence of ventricular ectopies and stopping episodes of the particular form of ventricular arrhythmia called torsade de pointes. Magnesium has also been described to have beneficial effects on the incidence of polymorphic ventricular tachycardia and supraventricular tachycardia. The implication of hypomagnesemia in the genesis of atrial fibrillation is well established; however, even if magnesium supplementation for rhythm control, cardioversion facility or cardioversion success/recurrence of AF after cardiac surgery and rate control during AF showed some benefit, it remains controversial. Although small randomised clinical trials showed a reduction in mortality when magnesium was administered to patients with acute myocardial infarction, the large randomised clinical trials failed to show any benefit of the administration of intravenous magnesium over placebo.

## 1. Introduction

Magnesium is a micronutrient, an alkaline earth metal occurring as a free cation (Mg^2+^). It is the second intracellular and the fourth extracellular electrolyte in concentration in humans. The adult human organism contains approximately 25 g of magnesium [1]. Up to 60% of the total Mg^2+^ is stored in the bone system, about 38–40% intracellular in the soft tissue and between 0.3 to 2% can be found in the blood serum [2]. Hypomagnesemia is defined as a total serum (Mg^2+^) < 0.65 mmol/L, is determined by measuring the total serum concentration by different methods and is often symptomatic [3]. The lower cut-off limit of normal magnesemia is set differently by various studies. However, a recently standardised agreed normal value of serum magnesium is 0.85–2.3 mg/dL [4]. There is an important limitation of these methods linked to the ability of Mg^2+^ to keep serum concentration within normal limits due to and at the cost of intracellular depletion [5]. There are currently no rapid and effective tests to assess the concentration of intracellular Mg^2+^, which specifically plays an essential role in both the heart and the whole body. However, there is a method that indirectly assesses the intracellular concentration of magnesium based on a reduction in excretion below 80% within 24 h after a loading dose [6].

Costello et al and later Rosanoff et al, described updated serum magnesium concentrations necessary for classification of magnesium status using adapted current reference intervals to describe symptomatic hypomagnesemia (<1.22 mg/dL), asymptomatic hypomagnesemia (1.22–1.82 mg/dL), chronic latent deficiency (1.82–2.06 mg/dL), normal range (2.06–2.33 mg/dL), asymptomatic hypermagnesemia (2.33–4.86 mg/dL) and symptomatic hypermagnesemia (>4.86 mg/dL). This classification introduces two new subclasses (asymptomatic hypomagnesemia and chronic latent deficiency). The new subclasses increase the accuracy of establishing magnesium status and correcting hypomagnesemia from early stages as it is known to date that chronic latent magnesium deficiency increases susceptibility to disease [4,7].

This review summarises the implications of magnesium in the cardiovascular processes focusing on its impact on the genesis and maintenance of supraventricular and ventricular cardiac arrhythmias. This paper’s main objective is to clarify the aspects related to the involvement of Mg^2+^ in cellular energy metabolism and other processes influencing the electrophysiological system of the heart. Identifying the implication of magnesium in mechanisms of arrhythmias and the potential causes of hypomagnesaemia, in light of the significant data identified so far, is the key to further management.

## 2. Causes of Hypomagnesaemia

Hypomagnesaemia can be a consequence of insufficient intake, redistribution from the extracellular to intracellular space and increased loss through the renal or gastrointestinal system [8,9].

Reduced magnesium intake can result from insufficient dietary consumption, alcohol dependence or parenteral nutrition that uses products with a poor magnesium content. Malnutrition, especially age-related, is most often associated with deficiency of macro-elements such as Mg^2+^, whose scarcity can lead to multiple implications, including a decline in physical performance and cardiovascular health issues represented mainly by hypertension and arrhythmias [10,11]. Redistribution of Mg^2+^ from the extracellular to the intracellular compartment can occur after surgical treatment of hyperparathyroidism (hungry bone syndrome) during the treatment of diabetic ketoacidosis, in refeeding syndrome, and during sympathetic stimulation (such as in alcohol withdrawal), in acute pancreatitis or other critical illnesses or postoperative states.

Magnesium renal loss due to tubulopathies is associated with hereditary conditions such as Gitelman syndrome, familial hypomagnesemia with hypercalciuria and nephrocalcinosis (FHHNC), Bartter syndrome, hypomagnesemia with secondary hypocalcemia (HSH), isolated recessive hypomagnesemia (IRH) with normocalcemia, isolated dominant hypomagnesemia (IDH) with hypocalciuria, and autosomal-dominant hypocalcemia with hypercalciuria (ADHH) [12]. Acquired causes include medications (loop and thiazide diuretics, aminoglycoside antibiotics, amphotericin B, immunosuppressive regimens: tacrolimus, cyclosporine, and chemotherapeutic agents such as cetuximab, panitumumab and cisplatin), and other pathologies: primary aldosteronism, chronic alcohol abuse (reversible tubular dysfunction), polyuria after urinary tract obstruction, or during acute tubular necrosis, hypercalcemia and hypophosphatemia [13].

Gastrointestinal losses can occur with vomiting, diarrhea, fistulas and nasogastric suction. Chronic malabsorption syndromes usually involve the small intestine and chronic use of protein pump inhibitors reduce the gastrointestinal absorption of magnesium [14], resulting in hypomagnesaemia.

It is worth mentioning that intracellular magnesium depletion can be associated with normal plasma values [15]. Yet, most patients with hypomagnesaemia will present with reduced total serum magnesium as well. More accurate measurements of Mg^2+^ can be obtained from lymphocytes and erythrocytes that have been correlated to intramyocardial muscle magnesium levels. Samples from buccal tissues can also be used to measure magnesium levels, though only available in the USA; however, such testing is reserved for research [16,17].

## 3. Cardiovascular Importance of Magnesium

The clinical implications of hypomagnesemia in the appropriate functioning of the cardiovascular system have been demonstrated over time. At the level of the vessels, the deficit of Mg^2+^ can promote increased vulnerability to oxygen-derived free radicals, altering the endothelial function and contributing to the genesis of atherosclerotic plaque formation [18]. An animal model study showed a reduction in infarct size associated with early reperfusion and magnesium administration. Furthermore, additional animal research highlights that Mg^2+^ supplementation decreases the myocardial infarct size when administered early, before reperfusion therapy, which is rather due to a direct cellular effect than to the regional myocardial blood reflow [19]. Although some small randomised clinical trials demonstrated a remarkable reduction in mortality when magnesium was administered to high-risk patients with acute myocardial infarction, three other large-scale randomised clinical trials (LIMIT 2, ISIS-4 and MAGIC trials) failed to show any benefit of the administration of intravenous magnesium over placebo. LIMIT 2 and ISIS-4 described different results, casting a shadow of doubt on the importance of magnesium in acute myocardial infarction; while in LIMIT 2 magnesium sulphate administration showed benefits both in patients who underwent thrombolysis and in those who did not, in ISIS-4 meta-analysis the evidence of magnesium supplementation benefit was absent in both groups [20,21,22]. The MAGIC trial assessed the influence of magnesium therapy on mortality in high-risk patients with acute myocardial infarction. Magnesium supplementation in the MAGIC trial aimed to prevent reperfusion injury; however, it did not demonstrate any benefit on 30-day mortality [21].

Besides ion disturbance and a large variety of arrhythmias, low Mg^2+^ is associated with increased vascular tone aggravating arterial hypertension [23], and apparently can affect valves and the course of heart failure (Table 1).

Magnesium strongly affects the myocyte’s homeostasis and maintains a normal action potential. Its use as an adjunctive therapy has demonstrated its importance in electrolyte balance in clinical practice. Recent data confirm a degree of interdependence between hypomagnesemia and cardiac arrhythmias.

Changes in the surface electrocardiograms (ECG) associated with hypomagnesemia have been described over time, dependent and varying according to magnesium levels: ST segment depression, peaked and tall or flattened T waves, shortening of the PR interval and QTc, lowering of the QRS voltage, increasing in the QRS duration and the presence of U waves. In a study by Yang et al., isolated hypomagnesemia was found to be associated with increased dispersion of ventricular repolarization evaluated by Tpe /QT ratio (T peak-to-end interval (Tpe), QT-peak interval standard measurement from the beginning of the QRS complex to the peak of the T-wave (QTp)) [24].

Cardiac arrhythmias have various causes and can be associated with a macroscopic structurally normal heart. Frequent examples of these types of causes are inherited conditions, as in the case of the accessory pathway-mediated tachycardias or the normal variant of nodal conduction duality characterized by the presence of a fast and a slow pathway with modified electrophysiologic properties predisposing to atrioventricular nodal reentrant tachycardia. The rest of the known arrhythmias are associated mostly with changes in the heart’s structure found in many pathological conditions and are represented mainly by atrial fibrillation, premature atrial contractions, premature ventricular contractions, ventricular tachycardia and ventricular fibrillation.

Hypomagnesemia has many causes and seems to be linked to disturbances in energy metabolism, ion channel exchanges, action potential alteration and myocardial cell electrical instability, all leading to different types of arrhythmia.

The implication of hypomagnesemia in the genesis or potentiation of cardiac arrhythmias has several mechanisms (Table 2). The first described mechanism is the inadequate effect of low-concentration magnesium against calcium in the atrioventricular node (AV node), as magnesium is known to be the natural Ca2+ antagonist.

Another aspect of arrhythmogenesis refers to different types of Mg^2+^ dependent K+ currents playing an essential role against the excessive prolongation of myocardial action potential and having a protecting role against arrhythmia [26]. The mechanisms behind hypomagnesemia-associated hypokalaemia seem to be linked to the alteration of Na-K-ATPase responsible for a decrease in the cellular uptake of K+ and increased renal K+ excretion due to the inhibition of the Na+/K+-ATPase. Low magnesium causes impairment of the function of Na+/K+-ATPase, which is Mg^2+^ regulated and increases intracellular sodium and reduces intracellular potassium levels, creating a less negative resting membrane potential. Magnesium deficiency can also lead to hypokalaemia and hypocalcemia, making the role of establishing the link between pure hypomagnesemia and cardiac arrhythmias somehow difficult. There is a strong relationship between Mg^2+^ and K+; both cations are integral for decreasing cell excitability and stabilizing membrane potentials. Half of the significant hypokalemias seem to be associated with magnesium deficiency; the co-administration of magnesium thus is essential in correcting serum potassium (K+) concentrations [27]. Enhanced gastrointestinal and renal excretion of K+ can also lead to hypokalaemia; however, it was shown that in patients with both hypomagnesemia and hypokalaemia, the repletion of Mg^2+^ alone also increased the serum K+ level [4,28].

The cellular mechanism behind the reduction of K+ excretion induced by Mg^2+^ repletion is linked to the effect on the ROMK. This inward-rectifying K+ current helps flow the potassium ion inside the cells of the apical membrane of the distal tubule. Intracellular magnesium binding to the channel’s pore causes a temporary block for efflux of K+. On the other hand, the influx of K+ displaces the channel-bent magnesium ion and the maximal needed quantity of K+ enters the cell. As a result, the arrhythmia precipitated by K depletion requires both K+ and Mg^2+^ repletion for termination [29].

Other hypomagnesemia-dependent effects are the induction of hypokalemia and hypocalcemia and the loss of the direct effect of the normal concentration of magnesium on preventing calcium overload in the myocardial cell. Hypomagnesemia also induces the elaboration of platelet activated factor (PAF), which raises the production of pro-inflammatory cytokines, thrombin and vasoactive mediators, creating and maintaining the inflammation of cardiac tissue that predisposes to arrhythmias [30].

A series of relationships implicating Mg^2+^ dependent energy metabolism have been described. Mitochondrial Mg^2+^ regulates reactions in mitochondrial energy metabolism, tricarboxylic acid (TCA) cycle, adenosine diphosphate (ADP)/adenosine triphosphate (ATP) translocation and the electron transport chain [31].

Cardiac function is heavily influenced by (Mg^2+^) and its various effects on vascular tone, peripheral vascular resistance, myocardial metabolism and Ca2+ homeostasis. Magnesium ions affect the myocyte’s ‘electrical properties by influencing the ion channels’ activity implicated in generating the cell’s action potential, particularly in phases 2 and 3. In phase 2, intracellular Ca2+ overload and therefore cell toxicity are prevented by (Mg^2+^)-dependent inhibition of the L-type Ca2+ channel. In phase 3, high (Mg^2+^) concentration seems to block the slow activating component of the rectifier K+ current and the inward rectifier K+ channel. Magnesium ion is a cofactor of the Na+/K+ ATPase pump, which is active in phase 4 of the action potential and necessary for resuming an appropriate resting membrane potential. Magnesium depletion can reduce the activity of the pump, resulting in partial depolarization [32]. In addition, (Mg^2+^) plays a role in myocardial excitation-contraction coupling by influencing the intracellular Ca2+ movement [33]. Magnesium binds calmodulin and troponin C and interacts with Ca2+ -transporting proteins such as the Na+/Ca2+ exchanger (NCX) and Ca2+-ATPase (SERCA). Concerning cardiac and vasculature function, Mg^2+^ has a role in vasodilation [34]. Furthermore, normal serum levels of Mg+2 reduce oxidative stress in endothelial cells and, consequently, endothelial inflammation [35,36].

A meta-analysis of six studies including 1550 patients admitted to intensive care units (ICU), showed that hypomagnesemia was associated with higher mortality [37]. On the other hand, recent data assessing the effect of Mg^2+^ on the mortality rate of 20,438 hospitalised patients suggested that both hypo- and hypermagnesemia were associated with increased mortality and, moreover, that hypermagnesemia should be considered a critical laboratory biomarker for mortality [38]. In-hospital acquired ionized hypomagnesemia is quite common, affecting up to 25% of hospitalised patients, and seems to be associated with a worse prognosis and an increase in in-hospital mortality [39]. One recent study evaluating changes in Mg^2+^ concentrations in patients with severe COVID-19 illness revealed that patients with in-hospital hypermagnesemia had a higher incidence of vasopressor-requiring cardiogenic shock and respiratory failure with the need for ventilation and acute kidney failure necessitating haemodialysis and increased mortality [40,41]. In addition, hypomagnesemia might be associated with severe arrhythmias in patients with septic shock and systemic inflammatory response syndrome. At the same time, high normal serum magnesium and hypermagnesemia seem to both be independent predictors of mortality in patients admitted to ICU for acute myocardial infarction [42,43]. Arrhythmia is the primary mechanism of death associated with hypomagnesemia.

### 3.1. Atrial Fibrillation

Atrial fibrillation is the most frequent arrhythmia in adults, with a high lifetime risk of 1:3 in persons of European ancestry with an index age of 55 [44]. Postoperative atrial fibrillation is defined as the onset of arrhythmia in the immediate period after surgery. It is the most relevant arrhythmia in patients undergoing heart surgery, but it also appears after non-cardiac surgery (10–30%) and vascular and colorectal surgery (5–10%). Perioperative AF is associated with increased mortality, morbidity and in-hospital stays [45]. Recent research showed that amiodarone and beta-blockers effectively reduce the atrial fibrillation burden and the combination of both seems to have even more benefits [46,47].

Adjunctive therapy with magnesium showed to be helpful in further reducing the incidence of atrial fibrillation in patients with a magnesium deficit.

Concerning the relationship between hypomagnesaemia and the development of atrial fibrillation in patients with no baseline cardiovascular disease, a large sample size and long follow-up study (20 years) showed that subjects with lower serum magnesium (≤1.77 mg/dL) had an approximately 50% increased risk of atrial fibrillation compared to those with magnesium levels in the upper limit. However, it should be mentioned that only the baseline serum magnesium was obtained [48]. In patients performing an electrophysiological study after intravenous magnesium infusion, a change in the conduction properties of the atrioventricular node was observed, not only in the sense of its prolongation but also in the increase of intratarial, interatrial conduction time and atrial and ventricular refractoriness [32,49,50]. This finding can give a future research direction in controlling heart rate in patients with atrial fibrillation or other atrial arrhythmias with high ventricular response, in addition to standard antiarrhythmic therapy [51].

In a large study, mild and moderate hypomagnesemia was associated with a significantly increased incidence of atrial fibrillation over a 20-year follow-up [52].

Some studies suggest a correlation between a low preoperative intracellular magnesium concentration and an increased risk of developing postoperative atrial fibrillation (AF) in patients undergoing cardiac surgery [53]. Hypomagnesemia was also associated with a high risk of atrial fibrillation after cardiac surgery; nevertheless, the intraoperative administration of magnesium showed no benefit in the reduction of this arrhythmia [54]. Cardiopulmonary bypass is also frequently associated with low serum magnesium levels and an increased risk of developing cardiovascular surgery-related atrial fibrillation [32,55]. The incidence of perioperative atrial fibrillation associated with low Mg^2+^ was evaluated at 20 to 40% in several studies. It can also be attributed to the difference in the burden of comorbidities in each study population [56].

A meta-analysis of 35 studies investigated the effect of parenteral magnesium administration as a treatment or prophylaxis for arrhythmias in patients with cardiovascular surgery. The most frequent arrhythmia was atrial fibrillation, followed by ventricular arrhythmic events. Magnesium reduced atrial fibrillation episodes in the absence of significant adverse events; however, it failed to prove its benefit in other arrhythmias [57].

Several significant meta-analyses and trials used the randomization of patients to Mg^2+^ following the effects in a variety of situations, such as the treatment of acute AF vs. placebo, new-onset of AF in the emergency department, prevention of AF during cardiac surgery, before AF cardioversion with a no-treatment control group and pre-electrical cardioversion of AF; however, all the analyses had disappointing results. Other data showed a reduction in the ventricular rate during atrial fibrillation after magnesium administration [58,59,60,61,62,63].

A study by Khan et al. on more than 3500 participants evaluated the implications of hypomagnesemia on the incidence of atrial fibrillation in a longitudinal community-based cohort, showing a moderate association in patients without cardiovascular disease [64]. Moreover, other evidence-based data reveals an even stronger association between hypomagnesemia, heart failure and atrial fibrillation [65,66]. Magnesium deficit in heart failure results from a combination of factors such as anorexia, reduction of gastrointestinal absorption caused by intestinal congestion, consumption of Mg^2+^ associated with increased sympathetic activity, enhanced Mg^2+^ urinary excretion caused by secondary aldosteronism, and the use of loop diuretics [67].

The importance of preventing atrial fibrillation lies mainly in the unfavourable long-term prognosis of this arrhythmia, especially when it becomes permanent and accompanied by comorbidities. Even if 50–87% of the patients with atrial fibrillation are initially asymptomatic, they may develop various symptoms and a less favourable prognosis over time [44,68,69,70]. Recent data shows that progression to persistent or permanent atrial fibrillation is associated with increased cardiovascular events, hospitalization and death; however, it remains unclear if the adverse events are caused by the evolution of the primary cardiovascular diseases, atrial fibrillation itself, or both [71,72]. Magnesium supplementation might be an accessible way to reduce the burden of atrial fibrillation; nevertheless, further evidence is needed to structure and deepen knowledge in this regard.

### 3.2. Premature Atrial Contractions (PAC)

The number of premature atrial and ventricular contractions increases with age, with a prevalence of 50% in the general population [73]. A low magnesium level has been suggested to increase the number of premature atrial and ventricular contractions in healthy subjects. A small study has shown that a reduction in less than half the recommended dose of dietary magnesium is associated with a slight increase in myocardial excitability [74]. While this effect can be considered of low significance in the healthy subject, it warrants attention in patients with pathologies predisposing to supraventricular arrhythmia. A pilot randomised trial showed no reduction in the number of PACs after the administration of 400 mg of magnesium daily over 12 weeks, in 59 adults, compared to placebo [75]. On the other hand, a recent two-phased randomised study on the effect of magnesium supplementation for PAC and PVC showed that even if magnesium is not a curative treatment for arrhythmia, it was associated with a reduction in premature complexes and symptoms in both phases [73].

### 3.3. Supraventricular Tachycardia

One of the first systematic evaluations, a study by Lorenzo et al., showed differences in the myocardial electrical properties evaluated by the invasive electrophysiological study before and after the administration of 6 g, 4% intravenous magnesium in a small cohort of 10 patients. The results included several modifications after magnesium sulphate administration: a decrease in the sinus node function, the prolongation of the atrioventricular (AV) conduction time, a Wenkebach block in the AV node occurring at a larger S1S1 atrial stimulation interval, the increase of the refractory periods of the AV node, and the prolongation of the PR interval on surface ECG. The data provided by this study is important because it indirectly reflects the importance of magnesium in preventing arrhythmias involving the two structures mentioned, the sinoatrial node and the AV node (sinoatrial re-entry, inappropriate sinus tachycardia, atrioventricular nodal reentrant tachycardia, atrioventricular reciprocal tachycardia). Another double-blind, randomised, placebo-controlled dose-response study regarding changes in the atrioventricular node electrophysiological properties was provided by Christiansen et al., and showed that a maximal atrioventricular conduction time was obtained by the infusion of 5 mmol magnesium, which was not extended by additional dose escalation [76].

Stiles et al. observed that magnesium supplementation inhibited the conduction primarily over the slow pathway in patients with a dual morphology atrioventricular node, increasing the tachycardia cycle length during atrioventricular nodal reentrant tachycardia [77]. The atrial refractoriness also increased with magnesium administration [32,76].

### 3.4. Premature Ventricular Contractions

Premature ventricular and atrial contractions have an increased incidence with age, reaching up to 50% prevalence in the general population.

Low serum magnesium was linked to a greater incidence of PVCs and a higher risk of ventricular tachycardia and torsades de pointes in obese patients with diabetes and after surgery [78,79].

An RCT showed that oral magnesium supplementation improved the symptoms and the frequency of premature ventricular and supraventricular contractions in patients without underlying ischemic or structural heart disease [80].

While the role of hypomagnesaemia frequently associated with hypokalaemia has been documented in patients with ventricular tachycardia and an underlying cardiac condition [81], there is not much information available on the association between Mg^2+^ deficiency and ventricular arrhythmia in subjects with structurally normal hearts. A case report published in 2005 presented the situation of a 12-year-old patient with premature ventricular contractions and sustained ventricular tachycardia. While the cardiac evaluation for arrhythmia was normal, the Mg^2+^ serum level was low. Oral supplementation with magnesium effectively suppressed both PVCs and the episodes of ventricular tachycardia, documented over a follow-up of 5 years [82].

Low serum magnesium seems to be a strong predictor of cardiac mortality in patients with atherosclerosis risk or treated with drug-eluting stents for acute myocardial infarction [83,84].

Research data shows an exacerbation of ventricular arrhythmia in patients with heart failure NYHA II-IV associated with hypomagnesemia and a benefit from magnesium supplementation, decreasing the number of premature ventricular contractions isolated or organised into couplets and non-sustained ventricular tachycardias [81]. The study of Ceremuzynsky et al. demonstrated the association of hypomagnesemia with heart failure in a proportion of more than one-third of the included patients with a significant urinary Mg^2+^ loss in this specific population; the complex interplay between heart failure and hypomagnesemia was considered to be the cause of ventricular arrhythmic events improved after the infusion of magnesium sulphate [81]. Mg^2+^ regulates the passing of Na+, K+, and Ca2+ through the voltage-dependent ion channels, generating the action potential as previously described. Hypomagnesemia-induced disturbance of the ion circulation is a cause of repolarization disturbance with QT dispersion and prolongation, the primary substrate of the triggered ventricular arrhythmia known as “torsade de pointes” [85,86]. From the electrophysiological point of view, the mechanisms implicated in inducing and maintaining polymorphic ventricular tachycardia and “torsade de pointes” are premature ventricular contractions, early after depolarizations, variability of repolarization and intramural re-entry circuits [87]. The 2015 *ESC Guideline of Management of Patients with Ventricular Arrhythmias and the Prevention of Sudden Cardiac Death* indicates that hypomagnesemia is associated with ventricular arrhythmias and sudden death; the intravenous administration of magnesium sulphate has been demonstrated to help suppress ventricular arrhythmia even in the presence of a normal magnesemia [88].

The current knowledge regarding the implications of Mg^2+^ in the genesis of cardiac arrhythmias is quite limited, with a small number of prospective studies, a significant amount of contradictory data and a failure to highlight its benefit in certain situations. Future directions of research are needed to fill the gaps in the evidence. More extensive randomised, blinded approach studies are required to shed light on the importance of Mg^2+^ in the genesis of arrhythmias and the role this ion might play in preventing life-threatening arrhythmias in patients with and without cardiovascular diseases in the presence of different acute or chronic comorbidities.

### 3.5. Ventricular Tachycardia and Ventricular Fibrillation

The American Heart Association’s recommendations indicate adding magnesium sulphate to the therapeutic regimen used in the management of refractory ventricular fibrillation and torsade de pointes [89]. The 2022 ESC Guidelines for the management of patients with ventricular arrhythmias recommend intravenous magnesium supplementation for torsade de pointes and long QT syndrome even in the absence of hypomagnesemia [90]. Mg^2+^ reduces the early after depolarization by influencing the T and L-type calcium channels. This process inhibits the calcium intracellular influx associated with delayed after-depolarizations. Another antiarrhythmic effect of Mg^2+^ is exerted by promoting potassium intracellular influx, thus correcting the dispersion of ventricular repolarization without shortening the action potential duration. Magnesium sulphate administration decreases the reinitiation of torsade de pointes or other ventricular tachycardia rather than converting the ventricular arrhythmia; however, no randomised controlled studies are yet available for evaluating the benefit of giving magnesium in torsade. Potassium is an important adjunctive therapy together with magnesium sulphate in patients with ventricular dysrhythmias [91]. Optimal and increased magnesium concentrations lower the activity of the rapid inward current of I_Kr_ (delayed rectifier K+ channel) [32,92]. However, magnesium supplementation did not exert a significant effect on the His-Purkinje conduction time, nor did it prolong the duration of the QRS complex at rest [93].

Several studies evaluated the benefits of magnesium supplementation for ventricular tachycardia associated with digoxin therapy. Digoxin inhibits Na+/K+-ATPase, which increases the intracellular concentration of Na+ and Ca2+. Magnesium is an essential cofactor for Na+/K+-ATPase; hence, magnesium deficiency can cause a further increase in intracellular sodium while decreasing intracellular potassium [94].

Hypomagnesemia was found to be responsible for an increased risk of digoxin toxicity, which can trigger malignant ventricular arrhythmia in patients with normal digoxin and K+ levels [95,96].

Liu et al. described the acquired long QT syndrome (QTc ≥ 450 ms in males and ≥460 ms in females) associated with an increased risk for ventricular arrhythmia, linked to lowering of the potassium, magnesium and calcium, chlorine and natrium levels in patients with chronic kidney disease [91]. However, a review by Vandael et al. showed that hypomagnesemia might not be a risk factor for QT interval prolongation, though supplementing magnesium sulphate can reduce the incidence of torsade de pointes in the absence of QT interval modification [97].

The population at risk for ventricular arrhythmia with mild hypomagnesemia includes patients with myocardial ischemia, myocardial infarction, heart failure, post cardiopulmonary bypass and acute illness [98].

In cases of refractory ventricular arrhythmia/torsade de pointes associated with acquired long QT syndrome that are unresponsive to intravenous magnesium and the correction of precipitating factors, the control of the arrhythmia may be obtained with isoproterenol or temporary transvenous pacing [90].

## 4. The Treatment of Hypomagnesemia

Oral or enteral magnesium is the preferred route of administration in outpatient care, and is used for asymptomatic hypomagnesemia and gastrointestinal tolerance. In contrast, intravenous magnesium repletion is the preferred route of administration in symptomatic hypomagnesemia and/or gastrointestinal intolerance and inpatient care [99].

Magnesium supplementation in patients with hypomagnesemia takes three factors into account: kidney function, the presence or absence of hemodynamic stability, and the severity of symptoms. In the case of hemodynamically unstable patients, 1 to 2 g of magnesium sulfate can be infused over 15 min. In symptomatic stable patients with severe hypomagnesemia, 1 to 2 g of magnesium sulfate will be administered over one hour. Magnesium dosage for the adult patient is usually 4 to 8 g of magnesium sulfate, slowly infused over 12 to 24 h. In children, the agreed dose is 25 to 50 mg/kg over 60 s. The dose is recommended to be repeated every 5–15 min up to a maximum of 2 g [100]. After magnesium supplementation in both inpatient and outpatient care, magnesemia should be rechecked (whether in an inpatient or outpatient setting) to evaluate the effectiveness of the treatment, with a frequency depending on the arrhythmic events and/or symptoms [99]. However, caution is needed when interpreting magnesemia, because even if serum Mg^2+^ rises quickly after treatment, intracellular Mg^2+^ takes much longer to normalize. It has been shown that in patients with normal renal function, magnesium repletion should be continued for two days after magnesemia normalizes [100]. The most frequent adverse effect is flushing. Nausea and vomiting, drowsiness and hypotension can occur when administering higher doses.

To monitor the serum level of magnesium, several strategies have been proposed over time, both to ensure the effectiveness of supplementation and to prevent hypermagnesemia that could increase the risk of adverse cardiovascular events as well as all-cause mortality in patients with critical pathologies admitted to intensive care units [37,101,102,103]. A potential explanation of this observation is provided by the recent study conducted by Carillo et al. on 746 patients. This study shows that hypermagnesemia is directly proportional to the deterioration of renal function often associated with critical illness, and that hypermagnesemia is a significant predictor of cardiovascular events and also increases all-cause mortality [104]. Other causes of hypermagnesemia may be increased intake, compartment shift with hemolysis, Addison’s disease, lithium therapy, and hypothyroidism. The avoidance of hypermagnesemia is the reason why caution is required in supplementing Mg^2+^ in this specific population. The cardiovascular effects of hypermagnesemia derive partially from the property of magnesium to behave as a calcium blocker and also to cause the potassium channel block associated with repolarization disturbances responsible for bradycardia, hypotension, prolonged PR interval, QRS and QT interval duration, complete heart block, and worsening of heart failure [105]. The treatment options are the infusion of calcium salts or hemodialysis in extreme hypermagnesemia [35].

To date, monitoring magnesemia may be challenging, especially since there are no available standardized protocols for serum magnesium measurement in different patient populations or the general population. There are no evidence-based reference range values to date, but rather only a consensus-based cut-off value for defining hypomagnesemia, established at 0.85 mmol/L (2.07 mg/dL or 1.7 mEq/L) [4]. For serum measurement of Mg^2+^ concentrations, the blood samples should be accurately prepared to avoid hemolysis since the magnesium stored in the erythrocytes exceeds serum magnesium [106]. Anticoagulant contamination of the blood sample should also be avoided [107]. In patients with poor magnesium status associated with diabetes mellitus, heart failure, Crohn’s disease and asthma, as well as in healthy subjects, serum analysis remains the most common and convenient method of determining magnesium concentrations; however, the loading test, a time consuming and complex method, seems to be the gold standard for assessing Mg^2+^ status, but it has limitations in patients with kidney and intestinal diseases [107,108].

The frequency of the determination of serum magnesium still remains at the discretion of the personnel involved in the care of hospitalised patients or subjects monitored in outpatient settings. An important aspect of potential future research is the impact of the increase in body weight on the magnesium necessary ^80^. Future studies are needed to clarify the doses required to adjust the recommended daily dose of magnesium proportionally with increased body weight.

## 5. Conclusions

Magnesium is involved in multiple metabolic, enzymatic, and electrophysiological reactions and mechanisms which are indispensable or have additional benefits on the electrical stability of the ventricular myocardium. Although some studies reveal the association of hypomagnesemia with supraventricular and ventricular arrhythmias, this remains controversial and requires further research. However, the importance of intravenous magnesium sulphate administration has been shown to be beneficial in reversing “torsade de pointes”, decreasing the incidence of polymorphic ventricular tachycardias (other than the “torsade”) associated with structural heart disease and decreasing the incidence and the ventricular response during atrial fibrillation. The benefit of magnesium for the reduction of cardiac arrhythmias seems to rely on several mechanisms: the regulation of intracellular ionic pumps for K+, Na+ and Ca2+, the reduction of early delayed after-depolarization triggering life-threatening ventricular arrhythmias, the decreasing of PAF- related inflammatory cardiac response linked to arrhythmias, the prevention of shortening of the action potential duration, the increasing of the atrioventricular node conduction time and the atrial and ventricular refractoriness. Magnesium supplementation in cardiac arrhythmias constitutes an interesting and stringent direction to investigate in order to translate knowledge into standardized and effective practical clinical directions.

## Figures and Tables

**Table 1 biomedicines-10-02356-t001:** Clinical manifestations of hypomagnesemia potentially influencing the cardiovascular system [25].

Hypomagnesemia	Clinical Manifestation
VESSELS	Atherosclerosis
	Impairment on endothelial function
	Coronary artery disease
	Arterial hypertension
	Increases mortality and morbidity after acute myocardial infarction
	Increases vascular tone and resistance
ARRHYTHMIAS	Torsades de pointes
	Other ventricular tachycardias
	Atrial fibrillation
	PACs
	PVCs
	Supraventricular tachycardias
	Sinus tachycardia
VALVES	Mitral valve prolapse
HEART FAILURE	Increased morbidity and mortality in patients with heart failure
ION DISTURBANCE	Hypokalaemia
	Hypocalcemia

**Table 2 biomedicines-10-02356-t002:** Hypomagnesemia-induced arrhythmogenic mechanisms.

Hypomagnesemia-Dependent Arrhythmogenic Mechanisms
inadequate effect of low Mg^2+^ concentrations against calcium in the AV node and ventricular myocardium
low magnesium causes malfunction of Na+/K+-ATPase, consequently creating a less negative resting membrane potential by decreasing the concentration of intracellular K+ and increasing intracellular Na+
hypomagnesemia-dependent hypokalaemia and hypocalcemia
direct effect of magnesium on preventing calcium overload in the myocite
PAF- related inflammatory cardiac response linked to arrhythmias
disturbance of the action-potential

## Data Availability

Not applicable.

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
