# Peer review of "The Role of Hypomagnesemia in Cardiac Arrhythmias: A Clinical Perspective"

_biomedicines, 2022, doi:10.3390/biomedicines10102356_

Round 1

Reviewer 1 Report

This is a review article regarding hypomagnesemia in cardiac arrhythmias. The authors well summarized the impact of hypomagnesemia in various arrhythmias, such as atrial fibrillation and ventricular tachycardia/fibrillation. 

This is clinically important, and this reviewer considers that the authors have well written the review article. This reviewer has some comments as described below. 

Major comments:

1.     Page 3, lines 87-89. The authors mentioned the randomized clinical trials. This part should be described in more details.

2.     In this manuscript, the authors used “myocardial cell”. Is it “myocyte”? 

3.     Page 5, lines 158-169. The authors mentioned that hypermagnesemia is also associated with high mortality. In the treatment of hypomagnesemia section (page 8, lines 297-304), the authors described about the higher doses administration, and the reviewer wonders how often we should monitor the serum levels of magnesium during the treatment. The authors should add how to monitor in this section.  

Author Response

COVER LETTER

Response to Reviewer 1

Dear Reviewer, thank you for your kind comments and valuable recommendations, which we tried to respect the best we could.

Major Comments Responses:

  1. Page 3, lines 87-89. The authors mentioned randomised clinical trials. This part should be described in more detail.

Done. We referred to a series of small experimental randomised clinical trials that we explained in more detail and mentioned the lacking citations.

We also referred to 3 other randomised large-scale clinical trials in humans, for which we made more accurate descriptions and included the citations.

We kindly ask you to retrieve the changes in the modified text document.

  1. In this manuscript, the authors used “myocardial cell”. Is it “myocyte”?

Done. When writing myocardial cell, we refer to the myocyte. The syntagm “ myocardial cell” appeared 5 times in the article’s text; therefore, 2 times we replaced it with the term “myocyte”.

  1. Page 5, lines 158-169. The authors mentioned that hypermagnesemia is also associated with high mortality. In the treatment of hypomagnesemia section (page 8, lines 297-304), the authors described the higher doses administration, and the Reviewer wonders how often we should monitor the serum levels of magnesium during the treatment. The authors should add how to monitor in this section.

We have added recent information from the literature regarding monitoring the serum magnesium level. We also reminded of the importance of avoiding hypermagnesemia from the perspective of the associated risk of increased mortality among critically ill patients.

Reviewer 2 Report

Dr Negru and colleagues in their review article provide a comprehensive overview on the role of serum magnesium concentration in cardiac arrhythmias. The field is interesting. The topic is of interest mainly for cardiologists and, in particular, for those with a special interest in cardiac arrhythmias.

I have the following concerns.

Major points

1)     Physicians are used to think about hypomagnesemia as a life threatening condition, especially regarding potential cardiac arrhythmias. Some discussion about the causes and the consequences of hypermagnesemia should be provided.

2)     A complex interplay exists between hypomagnesemia, heart failure and cardiac arrhythmias. The interaction between magnesium and arrhythmias in this particular setting with potential prognostic implications should be discussed (doi: 10.1016/j.ijcard.2020.08.062; doi: 10.1055/s-0037-1602803)

3)     The clinical implications of the review should be strengthened. How should magnesemia be monitored in cardiologic patients? Should it be regularly monitored in outpatients visits?

Minor points

1)     I would replace the last sentence of the abstract. The current sentence seems inconclusive, in my view it is not appropriate for a review article

2)     Malnutrition is associated with poorer functional status (10.1016/j.jchf.2020.04.004). Please discuss if a magnesium deficit may contribute to this issue.

3)     The conclusion section should be shortened to 2-4 brief sentences.

4)     A brief discussion about how magnesium should be corrected in outpatients and inpatients should be provided.

Author Response

COVER LETTER

Response to Reviewer 2

Dear Reviewer, thank you for your kind suggestions and comments. We did our best to fix all the issues raised in the report optimally.

Major points

  • Physicians are used to think about hypomagnesemia as a life-threatening condition, especially regarding potential cardiac arrhythmias. Some discussion about the causes and the consequences of hypermagnesemia should be provided.

Done. A short overview of hypermagnesemia, including causes and consequences, was inserted on page 9.

  • A complex interplay exists between hypomagnesemia, heart failure and cardiac arrhythmias. The interaction between magnesium and arrhythmias in this particular setting with potential prognostic implications should be discussed (doi: 10.1016/j.ijcard.2020.08.062; doi: 10.1055/s-0037-1602803)

Done. A brief reference to the interplay of hypomagnesemia- premature ventricular contractions- heart failure was already mentioned at the very beginning of page 8.

Next, we included a short description of another study on the interaction of heart failure-hypomagnesaemia- complex ventricular arrhythmias (page 8).

We also described the interrelationship between heart failure - atrial fibrillation and hypomagnesemia, highlighting the mechanisms of hypomagnesemia in heart failure (end of page 6 and beginning of page 7).

  • The clinical implications of the review should be strengthened. How should magnesemia be monitored in cardiologic patients? Should it be regularly monitored in outpatient visits?

Done. Data revealing the clinical importance of magnesium on the cardiovascular system was enriched (page 3). However, monitoring magnesemia may be challenging, especially since there are no available standardized protocols for serum magnesium measurement in different patient populations or in the general population. Consensus statements information on the techniques of measuring magnesium status were introduced on page 9. Still, information about the frequency of monitoring is lacking, this aspect being also specified in the manuscript following the advice of the Reviewer.

Minor points

  • I would replace the last sentence of the abstract. The current sentence seems inconclusive, in my view, it is not appropriate for a review article.

Done.

  • Malnutrition is associated with poorer functional status (10.1016/j.jchf.2020.04.004). Please discuss if a magnesium deficit may contribute to this issue.

Done. Included, page 2

  • The conclusion section should be shortened to 2-4 brief sentences.

Done.

  • A brief discussion about how magnesium should be corrected in outpatients and inpatients should be provided.

The requested informations were inserted into the section “The treatment of hypomagnesemia” page 9.

Round 2

Reviewer 2 Report

Thank you to the authors for their efforts in addressing my comments. The quality of the manuscript improved, but some points may still be improved.

Major points

1)     Thank you for your reply. Very interesting point.

2)     Thank you for your reply. However, in order to strengthen the clinical message and highlight the importance to prevent atrial fibrillation, I would suggest to mention the adverse prognostic role of atrial fibrillation (doi: 10.1016/j.ijcard.2020.08.062; doi: 10.2174/0929867324666170727115642).

3)     Thank you for your kind reply.

Further suggestion: Please update the reference with the latest guidelines on ventricular arrhythmias (10.1093/eurheartj/ehac262)

Author Response

COVER LETTER

Response 2 to Reviewer 2

Major points

  • Thank you for your reply. Very interesting point.

We are happy to have clarified this point successfully.

  • Thank you for your reply. However, in order to strengthen the clinical message and highlight the importance to prevent atrial fibrillation, I would suggest to mention the adverse prognostic role of atrial fibrillation (doi: 10.1016/j.ijcard.2020.08.062; doi: 10.2174/0929867324666170727115642).

Thank you for your kind suggestions. The idea to strengthen the message of the burden of atrial fibrillation in the light of prevention is more than welcome and we appreciate it.

We are sure the articles you suggested are very interesting, with appliance for our review article.

However, we cannot access them because those are not open access articles, we need to purchase them and we do not actually have more budget for this purpose. We tried to accomplish the need to mention the prevention of atrial fibrillation in light of the adverse prognostic role of permanent atrial fibrillation, using the European Society of Cardiology Guidelines 2020 and other related studies as references instead.

We included the text on page 7 (last paragraph from Atrial fibrillation).

Maybe you could kindly consider this as a reasonable solution.

3)     Thank you for your kind reply.

Further suggestion: Please update the reference with the latest guidelines on ventricular arrhythmias (10.1093/eurheartj/ehac262)

Done. Thank you for another valuable recommendation. We included 2 paragraphs on pages 8 and 9, respectively, at the beginning and the end of the “Ventricular tachycardia and ventricular fibrillation” section.

Round 3

Reviewer 2 Report

Thank you for your efforts in addresing my comments. In my view, the quality of the manuscript improved. I have no further suggestions.